# *BnLPAT2* gene regulates oil accumulation in *Brassica napus* by modulating linoleic and linolenic acid levels in seeds

**Luyao Huang[1], Zhiqiang Liao[1], Yujing Zou[1], Yong Liu[2], Huihui Wang[2], Leping Zou[1], Sun Liang[1], Shan Tong[1], Yu Kang[3], Tuo Chen[4], Xinghua Xiong[3]\*, Man Xing[5]\***

**1** Yichun Academy of Sciences, Yichun, Jiangxi, China, **2** College of Agronomy, Hunan Agricultural University, Changsha, Hunan, China, **3** Department of Bioengineering, Huaihan Normal University, Huainan, Anhui, China, **4** Anxiang County Agricultural and Rural Affairs Bureau, Anxiang, Hunan, China, **5** College of Agronomy and Biotechnology, Southwest University, Chongqing, China

\* xingman1897@163.com (MX); xiongene@hunau.com.cn (XX)

## Abstract

Lysophosphatidate acyltransferase (LPAT) catalyzes the conversion of lysophosphatidic acid to phosphatidic acid, a key step in lipid biosynthesis. This study cloned four *LPAT2* genes from *Brassica napus*: *BnLPAT2-A04, A07, A09,* and *C08*. Functional analysis using bioinformatics, qRT-PCR (Quantitative Reverse Transcription Polymerase Chain Reaction), CRISPR/Cas9 (Clustered Regularly Interspaced Short Palindromic Repeats/CRISPR-associated protein 9), overexpression, and transcriptome sequencing revealed that these genes encode proteins containing the conserved PLN02380 domain. *BnLPAT2-A07/A09/C08* showed strong conservation with Arabidopsis *AtLPAT2*. Promoter analysis revealed multiple cis-elements related to stress, light, and phytohormone responses, with the *BnLPAT2-A09/C08* promoters containing the most diverse cis-elements. Expression analysis showed that *BnLPAT2-A07/C08* was highly expressed in various tissues, with *BnLPAT2-A07* peaking during seed development. Overexpression of these genes increased seed oil content and the proportion of C18:2/C18:3 fatty acids, with *BnLPAT2-A07* achieving an increase in oil content ranging from 4.46% to 6.44%. Gene knockout reduced oil content by 7.5% and affected fatty acid accumulation. Transcriptome sequencing analysis suggested that the *BnLPAT2* genes promote the production of long-chain fatty acids, such as Linoleic acid (C18:2) and Linolenic acid (C18:3), through biological processes, including fatty acid biosynthesis, very long-chain fatty acid biosynthesis, and very long-chain fatty acid metabolism, thereby improving seed oil content. This study provides valuable insights into lipid metabolism and offers a theoretical foundation for improving oil content and fatty acid composition in *B. napus*.

## Introduction

Increasing the oil content of oil crop seeds is crucial for enhancing plant oil production. In the Kennedy pathway of plant oil biosynthesis, glycerol-3-phosphate acyltransferase (GPAT) [1–4], lysophosphatidic acid acyltransferase (LPAT) [5,6], and diacylglycerol acyltransferase

**Data availability statement:** All relevant data are within the manuscript and its Supporting Information files.

**Funding:** This work was supported by the Natural Science Foundation of China (32401904), the Natural Science Foundation of Chongqing (CSTB2024NSCQ-MSX0906), Special fund for guiding city and county science and technology development of Jiangxi Province, and the National Key Research and Development Program of China (2017YFD0101700).

**Competing interests:** The authors have declared that no competing interests exist.

(DGAT) [7,8] are key enzymes involved in triacylglycerol (TAG) formation in plant seeds. GPAT catalyzes the acylation of the sn-1 position of glycerol by acyl-CoA, forming lysophosphatidic acid (LPA) [9]. LPAT catalyzes the acylation of the sn-2 position of glycerol, producing phosphatidic acid (PA) [10]. DGAT regulates the final step of triacylglycerol (TAG) synthesis by catalyzing the formation of triacylglycerol from diacylglycerol [11,12]. LPAT uses lysophosphatidic acid as a substrate to produce phosphatidic acid and is a key enzyme in lipid synthesis in living organisms.

In *Arabidopsis thaliana*, LPAT is encoded by five genes: *AtLPAT1*, *AtLPAT2*, *AtLPAT3*, *AtLPAT4*, and *AtLPAT5*. *AtLPAT1* is associated with embryonic development [13], *AtLPAT2* is localized to the endoplasmic reticulum and affects female gametophyte development [14], while *AtLPAT3* is related to another development [15]. Research on *AtLPAT4* and *AtLPAT5* is relatively scarce. Based on subcellular localization, the LPAT family is classified into two main types: plastid-type and endoplasmic reticulum-type. The ER-type LPATs are further divided into two categories: type A and type B [16]. Type A LPATs are expressed in all plant tissues and exhibit a preference for C18:1-CoA [17]. In contrast, type B LPATs are primarily expressed in tissues that accumulate oils and show a preference for saturated or unsaturated acyl substrates [18]. In castor (*Ricinus communis*), three LPAT-A genes (*RcLPAT2*, *RcLPAT3*, *RcLPAT4*) have been reported. Their gene expression varies during different stages of seed development, with *RcLPAT2* showing the highest relative expression [19] and a stronger preference for C18:1-CoA [20]. This indicates that although LPATs from different species exhibit varying preferences for fatty acids, they all influence fatty acid and TAG synthesis. This highlights the importance of *LPAT* as a key gene in lipid metabolism, with significant research value in regulating seed oil content.

Research on the *LPAT2* gene has made significant progress in *Arabidopsis thaliana* [21,22], *Cuphea* [16], *Ricinus communis* [5], *Camelina sativa* [23,24], *Consolida ajacis* [25], *Perilla frutescens* [26], and rapeseed [27,28]. Woodfield overexpressed the *BnLPAT2* gene in rapeseed and found a significant increase in triacylglycerol content in the seeds [28]. Zhang [27] used CRISPR/Cas9 gene-editing technology to perform targeted editing of multiple homologous copies of *BnLPAT2* and *BnLPAT5* in rapeseed. The results showed that oil content decreased by 32% and 29% in *BnLPAT2* and *BnLPAT5* edited lines, respectively, confirming a close relationship between these genes and silique development and oil accumulation in rapeseed. Introducing yeast *ScLPAT* into rapeseed increased both seed oil content and crop yield [29]. Similarly, Chen [30] introduced the peanut *AhLPAT2* gene into *Arabidopsis*, resulting in increased seed weight, oil content, total fatty acids, and unsaturated fatty acids. Castor *RcLPAT2* enhanced oleic acid content in *Lesquerella* seeds [31]. Overexpression of *Camelina sativa CsaLPAT2* significantly increased the long-chain fatty acid content in triacylglycerol, slightly enlarged seed size, significantly increased total phospholipid content, and raised the proportion of phosphatidic acid molecules containing long-chain fatty acids to 45%, which was 2.8 times higher than in the wild type [23].

These studies demonstrate a strong association between *LPAT2* expression seed oil content and fatty acid composition. Recent research findings indicate that introducing the rapeseed *BnLPAT2* gene into Arabidopsis can increase seed oil content and significantly enhance linolenic acid content [27]. Although numerous studies have reported that heterologous expression of the *LPAT2* gene can increase the linolenic acid content and long-chain fatty acid content in oils [23,27,30,31], the underlying mechanism has not yet been elucidated. Consequently, *LPAT2* has become a focal point for research on improving oil composition and increasing oil content. In this study, four *LPAT2* genes from *B. napus* were cloned and named *BnLPAT2-A04* (*BnaA04G0028300ZS*), *BnLPAT2-A07* (*BnaA07G0200200ZS*), *BnLPAT2-A09* (*BnaA09G0530700ZS*), and *BnLPAT2-C08* (*BnaC08G0376200ZS*). Using *Arabidopsis* and

rapeseed lines with *LPAT2* overexpression and CRISPR/Cas9 knockouts, lipidomic and transcriptome analyses of seeds were conducted to elucidate further the molecular mechanisms by which *BnLPAT2* regulates seed oil content and fatty acid accumulation. Thus, in-depth research on *BnLPAT2* will enhance our understanding of fatty acid biosynthesis mechanisms in rapeseed seeds, providing valuable insights for improving oil accumulation and modifying fatty acid composition in oilseed crops.

## Materials and methods

### Materials

The *B. napus* ZS6 used in this study was provided by the rapeseed team at Southwest University, while the *B. napus* XY15 was obtained from the Oil Crops Research Institute of Hunan Agricultural University. ZS6 and XY15 were planted in the experimental fields at Southwest University. Seeds from $T_2$ homozygous transgenic lines were collected for oil content and fatty acid analysis. The $T_2$ generation transgenic rapeseed and its controls were cultivated in an isolated greenhouse under conditions consistent with field planting, except for the isolation. We measured the expression levels of the *BnLPAT2* gene in different tissue parts of ZS6 and XY15. During the flowering stage, roots, stems, leaves, flower buds, and flowers were collected, immediately flash-frozen in liquid nitrogen, and stored at −80°C for later use. Siliques at 7, 14, 21, 28, 35, and 42 days after pollination were sampled, and seeds were carefully separated from the silique walls with forceps. The seeds were then flash-frozen in liquid nitrogen and stored at -80°C. The *Arabidopsis thaliana* used in this study was the Columbia wild type, which was preserved in our laboratory.

### Cloning and bioinformatics analysis of four *BnLPAT2* genes in *B. napus*

Retrieve the ID of *AtLPAT2* (AT3G57650) from the TAIR database (https://www.arabidopsis.org/) and copy the Arabidopsis gene ID to the BnTIR database (https://yanglab.hzau.edu.cn/BnTIR) to identify the four homologous copies of the *BnLPAT2* gene in *B. napus*. Download the mRNA sequences. The primer design for the *BnLPAT2* gene is shown in S1 Table. Use the NCBI database (https://www.ncbi.nlm.nih.gov/) with the ORF Finder and CD-search tools to analyze the open reading frame and protein domains of *BnGPAT9* in *B. napus*. Predict the secondary structure using the SOPMA software (http://npsa-pbil.ibcp.fr/cgi-bin/). Analyze the protein's molecular weight, isoelectric point, and other basic physicochemical properties using the ExPASy tool (https://web.expasy.org/protparam/). Analyze cis-regulatory elements using the PlantCARE database (http://bioinformatics.psb.ugent.be/webtools/plantcare/html/) and TBtools software. Use the MEME program (http://meme-suite.org/tools/meme) to analyze conserved motifs in the protein sequence. Using the *BnLPAT2* amino acid sequence as the reference, perform a Blastp search to identify homologous amino acid sequences in other species and construct a phylogenetic tree using MEGA6.06 software.

### Plant expression vector construction and genetic transformation

Upstream primers for cloning the *BnLPAT2* gene were designed to include a BamHI (GGATCC) restriction enzyme site at the 5' end, and downstream primers included a SacI (GAGCTC) restriction enzyme site at the 5' end. The primers were used to ligate the gene into the plant binary expression vector pBI121 to construct a *CaMV35S* promoter driven overexpression vector. The *Napin* promoter sequence was retrieved from the *B. napus* database, and primers were designed based on the sequence as follows: *Napin-F*: AAGCTTGTTCAAG-CGAATGGCATACCG; *Napin-R*: GGATCCTGTTTGTATTGATGAGTTTTGG (underlined regions indicate restriction enzyme sites). The *Napin* promoter replaced the *CaMV35S*

promoter in the pBI121 vector to construct a *Napin* promoter driven overexpression vector. The sgRNAs for *BnLPAT2-A07* were designed using the CRISPR Design Tools website (https://www.synthego.com/products/bioinformatics/crispr-design-tool), yielding the sequences S1 (TGTCTGGATCGTTGACTGGTGG) and S2 (TCACCGAAGTGATATT-GATTGG). The construction of the editing vector and the detection of CRISPR/Cas9 editing in rapeseed were carried out according to the experimental methods described in the CRISPR/Cas9 toolkit for plant multiplex genome editing by Xing [32]. The transformation was performed using the *Agrobacterium tumefaciens* GV3101 mediated floral dip method [33] for *Arabidopsis thaliana* and the hypocotyl transformation method [34] for *B. napus*. The harvested Arabidopsis seeds after transformation are spread on 1/2 MS solid medium containing kanamycin to select $T_0$ plants. The $T_0$ plants were then grown in a growth chamber, and after harvesting $T_0$ seeds, further screening was performed until homozygous lines were obtained. Seeds from the $T_2$ generation of *Arabidopsis* were harvested to measure oil content and fatty acid composition. The screening and identification of transgenic plants were carried out in a growth chamber with a temperature of 24°C and a photoperiod of 16 hours light and 8 hours dark. The primers for detecting transgenic positive lines of knock-out and overexpression rapeseed are shown in S2 Table.

## Determination of seed oil content and fatty acid composition

The oil content and fatty acid content of *Arabidopsis thaliana* and rapeseed seeds were determined using a gas chromatograph. The steps for fatty acid extraction were as follows:

1. Weighed Seeds: 10 mg of seed samples were weighed.

2. Added Reagent: 2 mL of a methanol solution containing 2.5% sulfuric acid (with 0.01% BHT) was added.

3. Added Internal Standard: 100 μL of 16.2 μmol/mL C17:0 fatty acid was added as an internal standard.

4. Processed Samples: For rapeseed samples, seeds were gently crushed using a glass rod (Arabidopsis seeds did not require this step).

5. Heated in Water Bath: Samples were heated in an 85°C water bath for 2 hours, with the lid checked every 10–20 minutes to ensure it was secure and not leaking.

6. Cooled Samples: After heating, the samples were cooled to room temperature.

7. Performed Liquid-Liquid Extraction: 2 mL of ultrapure water (ddH₂O) and 2 mL of n-hexane were added, and the mixture was vortexed thoroughly.

8. Centrifuged Samples: The samples were centrifuged at 1000 rpm for 10 minutes.

9. Collected Supernatant: Approximately 1 mL of the supernatant was collected into sample vials.

10. Injected Samples: 1 μL of the sample was injected into the gas chromatograph with a split ratio of 2:1–10:1.

The seed oil content was calculated using the following formula:
Seed oil content (%) = [mass of internal standard FA (C17:0)/ proportion of internal standard FA (C17:0) - mass of internal standard FA (C17:0)]/ seed mass.
Gas chromatographic measurement method: The gas chromatograph model used was the Agilent 6850 Network GC system, equipped with an Agilent Series Auto Sampler for injections of 1μL each time. The initial temperature of the chromatography column was 180°C,

gradually heated to 225°C, with a heating duration of 7 minutes. Nitrogen was used as the carrier gas for the column, with a flow rate of 1 mL/min. After injection and passage through the column, the OpenLAB analysis software provided by Agilent was used to obtain the gas chromatography measurement results of fatty acid methyl esters in each sample. This study employs the internal standard method to determine the absolute content of each fatty acid component in the samples, using heptadecanoic acid (analytical grade) as the internal standard, added in equal amounts to each sample before methylation.

### RNA extraction and qRT-PCR analysis

During the flowering period, the roots, stems, leaves, flowers, flower buds, and seeds at different developmental stages (7 days, 14 days, 21 days, 28 days, 35 days, and 42 days after pollination) of rapeseed varieties ZS6 and XY11 were collected. These samples were quickly frozen in liquid nitrogen and thoroughly ground. Total RNA was extracted using a Plant Total RNA Rapid Extraction Kit (Sangon Biotech, Shanghai, China) following the manufacturer's instructions. cDNA was synthesized using the One-Step gDNA Removal and cDNA Synthesis SuperMix (TransGen Biotech, Beijing, China). RT-PCR amplification was performed using the reverse-transcribed cDNA as a template and specific primers. The PCR reaction system included 1 μL of template, 1 μL of forward primer, 1 μL of reverse primer, 10 μL of 2×Phanta Flash Master Mix, and 7 μL of ddH₂O. The PCR conditions were as follows: 94°C for 3 min; 35 cycles of 94°C for 30 s, 60°C for 30 s, 72°C for 90 s; and a final extension at 72°C for 10 min. The target bands were detected via 1% agarose gel electrophoresis. For qRT-PCR analysis, *B. napus Actin2.1* served as the reference gene, with primers listed in S3 Table. Relative expression levels were calculated using the 2^(-ΔΔCt) method as described by Livak [35].

### Transcriptome sequencing analysis of transgenic rapeseed

At 30 days after pollination, siliques were collected, and seeds were removed, flash frozen in liquid nitrogen, and then stored at −80ºC for further analysis. Each treatment included three independent biological replicates. RNA was extracted from the seeds using the TRIzol method, followed by assessments of RNA purity and integrity. Libraries were constructed using the NEBNext Ultra RNA Library Prep Kit. After library construction, initial quantification was performed with a Qubit 2.0 Fluorometer, and the library was diluted to 1.5 ng μL⁻¹. The insert size of the library was then verified using an Agilent 2100 Bioanalyzer. Once the insert size met expectations, the effective concentration of the library was accurately quantified using qRT-PCR, ensuring a concentration above 2 nmol L⁻¹ for quality assurance. Sequencing was performed using the Illumina high-throughput sequencing platform (HiSeq2000). Differentially expressed genes (DEGs) were identified with the criteria |log₂(Fold Change)| ≥ 1 and q < 0.05. Functional annotation and analysis of DEGs were conducted using the GO (Gene Ontology) and KEGG (Kyoto Encyclopedia of Genes and Genomes) databases. Using the *zs11_hzau* (http://cbi.hzau.edu.cn/rape/download_ext/zs11.genome.fa) as the reference genome, the transcriptome data for the six homologous copies of the *BnLPAT2* gene were obtained from the BnTIR database (https://yanglab.hzau.edu.cn/BnTIR).

## Results

### Cloning and bioinformatics analysis of the *BnLPAT2*

Total RNA was extracted from XY15 seeds two weeks after pollination (Fig 1A). Using reverse transcribed cDNA as a template, PCR amplification produced four target bands, as shown in Fig 1B. The target fragments were recovered, ligated into the pUCm-T vector, and transformed into DH5α (*Escherichia coli*). After sequencing and alignment, four

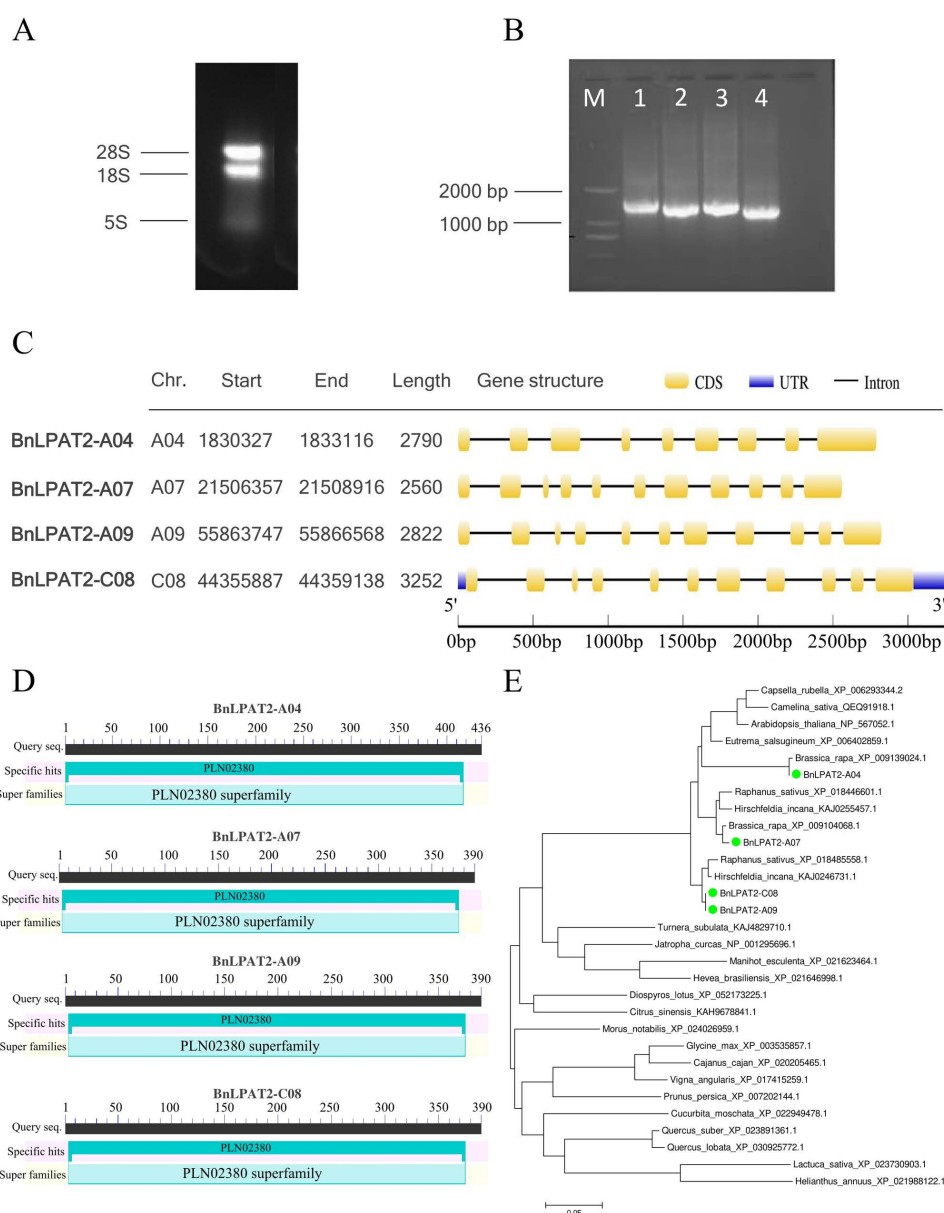

**Fig 1. Cloning of four *BnLPAT2* genes and phylogenetic analysis of their encoded proteins.** (A) Total RNA from *B. napus* XY15. (B) PCR results of *BnLPAT2* genes. M: 2K DNA Marker; 1: *BnLPAT2-C08*; 2: *BnLPAT2-A09*; 3: *BnLPAT2-A07*; 4: *BnLPAT2-A04*. (C) Chromosomal localization and gene structure of *BnLPAT2* genes. (D) Conserved domains of BnLPAT2. (E) Phylogenetic tree for BnLPAT2 proteins from different plant species.

full-length CDS sequences were cloned and named *BnLPAT2-A04* (1155 bp), *BnLPAT2-A07* (1173 bp), *BnLPAT2-A09* (1173 bp), and *BnLPAT2-C08* (1173 bp), encoding proteins of 384, 390, 390, and 390 amino acids, respectively. Gene structure analysis revealed that only *BnLPAT2-C08* contained 5'UTR and 3'UTR regions, while the other three genes displayed similar exon-intron distribution patterns (Fig 1C). The molecular weights of the *BnLPAT2* proteins ranged from 43.2 to 43.7 kDa. All were identified as unstable proteins with isoelectric points above 8, classifying them as basic proteins (S4 Table). Secondary structure predictions indicated that BnLPAT2 proteins consist of random coils, α-helices, extended strands, and

β-sheets, with α-helices being the most predominant, accounting for 46.15% (BnLPAT2-C08), 47.68% (BnLPAT2-A09), 48.21% (BnLPAT2-A07), and 45.57% (BnLPAT2-A04), respectively. β-sheets were the least abundant, at approximately 5% (S5 Table).

Protein domain prediction results revealed that all four BnLPAT2 proteins contain the conserved PLN02380 domain, classifying them within the PLN02380 superfamily (Fig 1D). Phylogenetic analysis showed that BnLPAT2-A09/C08 are closely related to *Hirschfeldia incana* and *Raphanus sativus*, whereas BnLPAT2-A04/A07 are more closely related to the LPAT2 proteins of *Brassica rapa* (Fig 1E). The conserved motifs of BnLPAT2-A07/A09/C08 and *Arabidopsis thaliana* AtLPAT2 proteins are highly conserved, all containing Motifs 1–10 with identical motif arrangements. However, the conserved motifs of the BnLPAT2-A04 protein differ, lacking the Motif 9 sequence (Fig 2A). Amino acid sequence alignment revealed that the BnLPAT2-A09/C08 sequences are identical, while BnLPAT2-A04 exhibits significant differences compared to the other three BnLPAT2 proteins, sharing only 77.83% similarity with AtLPAT2. In contrast, BnLPAT2-A07/A09/C08 share 91.82% and 98.04% similarity with AtLPAT2, respectively (Fig 2B).

## Analysis of Cis-acting elements in the promoter of *BnLPAT2* genes

To analyze the potential functions of the four *BnLPAT2* genes in rapeseed, the PlantCARE software was used to analyze the promoter sequences in the 2 kb upstream region of the *BnLPAT2* genes. As shown in Fig 3, a total of 51 types of functional cis-acting elements were identified in the promoter sequences of the *BnLPAT2* genes in rapeseed. These elements were categorized into five functional groups: Stress responsive elements, Light responsive elements Phytohormone regulation, Metabolism regulation, and other elements. Among them, light responsive elements were the most diverse, with G-box, BOX 4, and GT1-motif being the main types. Next were stress responsive elements, including MYB, MYC, ARE, and STRE. Additional light responsive elements included BOX 4, GT1-motif, and TCT-motif. Phytohormone regulation included those related to abscisic acid (ABRE), auxin (AuxR-core), methyl jasmonate (CGTCA-motif and TGACG-motif), and salicylic acid (TCA-motif). Metabolism regulation was primarily associated with carbon metabolism (AAGAA-motif) and arginine metabolism (O2-site). Moreover, the *BnLPAT2-C08* gene promoter sequence contained the most diverse types of cis-acting elements, followed by the *BnLPAT2-A09* gene, indicating that *BnLPAT2-C08/A09* in rapeseed have greater potential to participate in a broader range of physiological regulatory processes compared to other homologous genes.

## Expression of *BnLPAT2* genes in different tissues

Based on the transcriptome data reported in the BnTIR database, the expression levels of six homologous copies of the *BnLPAT2* gene across various tissues were analyzed using heat-map analysis (Fig 4A). The results showed that *BnLPAT2-C08/A09* exhibited high expression levels in roots, stems, leaves, flowers, siliques, and seeds, followed by *BnLPAT2-A07*, while *BnLPAT2-A04/C04* displayed the lowest expression levels in all tissues. Total RNA was extracted from roots, stems, leaves, flower buds, flowers, and seeds at different developmental stages of ZS6 and XY15, and cDNA was synthesized by reverse transcription for qRT-PCR analysis. Due to the high homology between *BnLPAT2-C08* and *BnLPAT2-A09*, only the sequence of *BnLPAT2-C08* was used to design specific primers for real-time quantitative PCR amplification.

We found that *BnLPAT2-A04/A07/C08* had similar expression patterns in roots, stems, leaves, and flowers of ZS6 and XY15, with *BnLPAT2-A07/C08* showing higher expression levels and *BnLPAT2-A04* showing very low expression. However, in XY15 flowers, *BnLPAT2-A04*

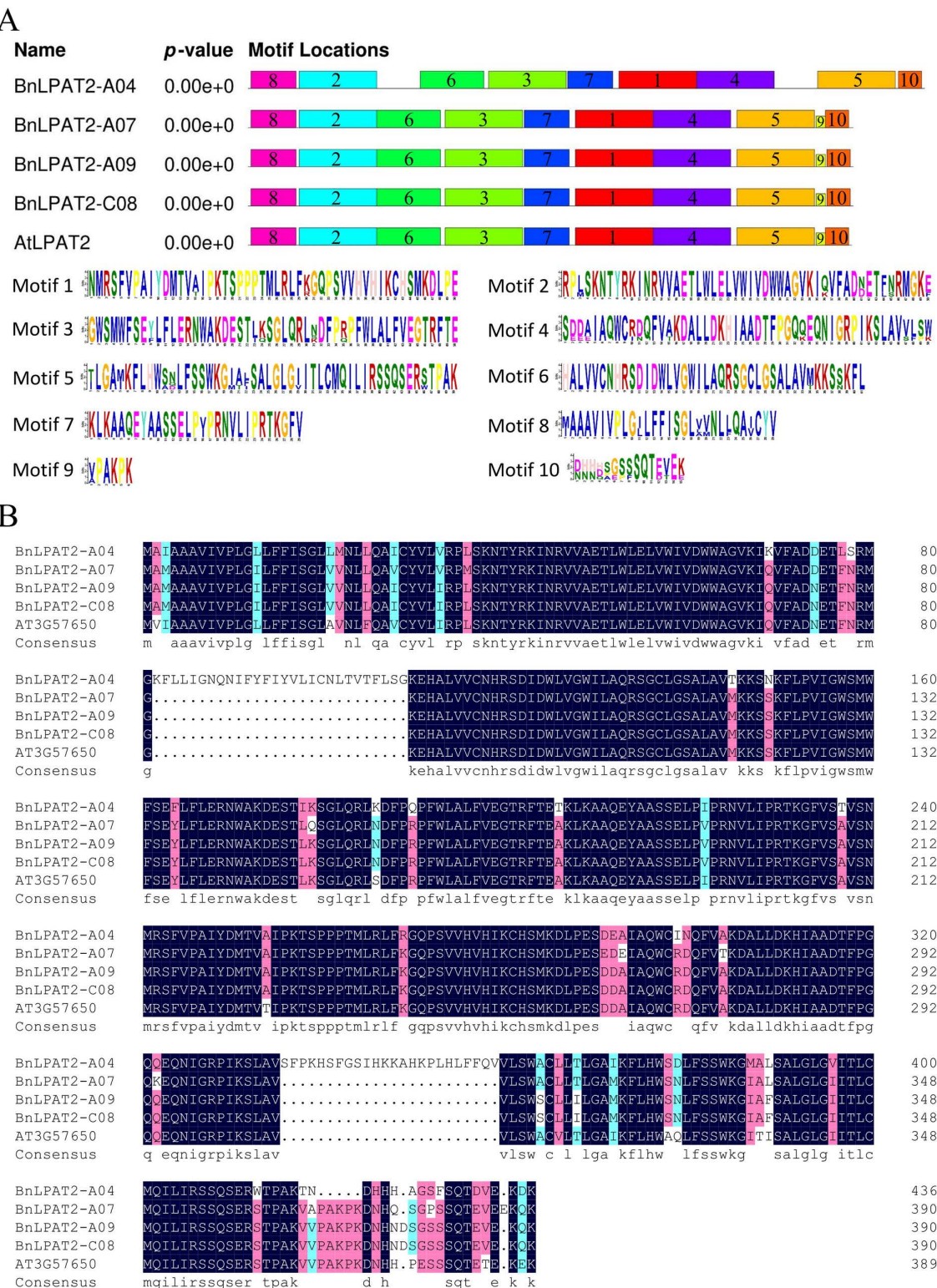

**Fig 2. Characterization of the *BnLPAT2* protein sequences.** (A) The analysis of conserved motifs in BnLPAT2 and AtLPAT2 proteins. (B) The multiple sequence alignment of BnLPAT2 and AtLPAT2 proteins.

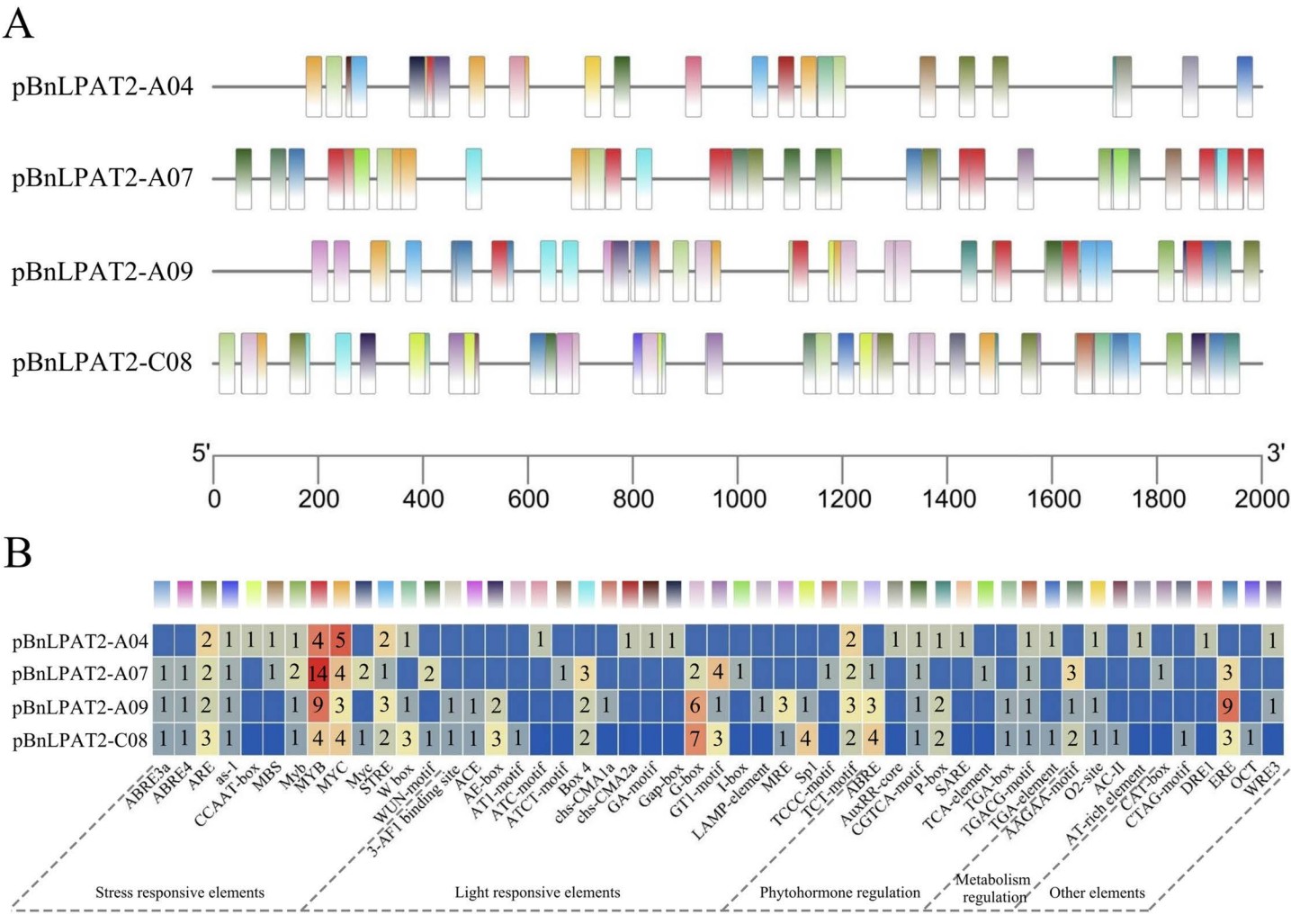

**Fig 3. Cis-elements detected in the promoter of the *BnLPAT2* genes.** (A) Promoter element distribution, different colors correspond to different elements in the figure below. (B) The heat map shows the number of promoter elements, and the blue square indicates that the elements couldn't be detected.

exhibited a higher expression level (Fig 4B and D). Additionally, *BnLPAT2-A04/A07/C08* displayed distinct expression patterns during seed development in ZS6 and XY15 (Fig 4C and E). *BnLPAT2-C08* showed high expression levels in both early and late stages of seed development in ZS6, while in XY15, its expression decreased as seed development progressed. The expression of *BnLPAT2-A04* differed significantly between the two varieties: in XY15, it maintained high expression levels during seed development, gradually decreasing as seed development progressed, whereas in ZS6, it exhibited relatively high expression levels only in the late stage of seed development. Notably, *BnLPAT2-A07* exhibited similar expression patterns during seed development in ZS6 and XY15, with higher expression levels detected only in the mid-to-late stages of seed development.

## *BnLPAT2* genes transformation in *Arabidopsis* and rapeseed and seed oil content analysis

We constructed overexpression vectors driven by the *CaMV35S* promoter for four *BnLPAT2* genes and genetically transformed *Arabidopsis*. We found that only the overexpression of

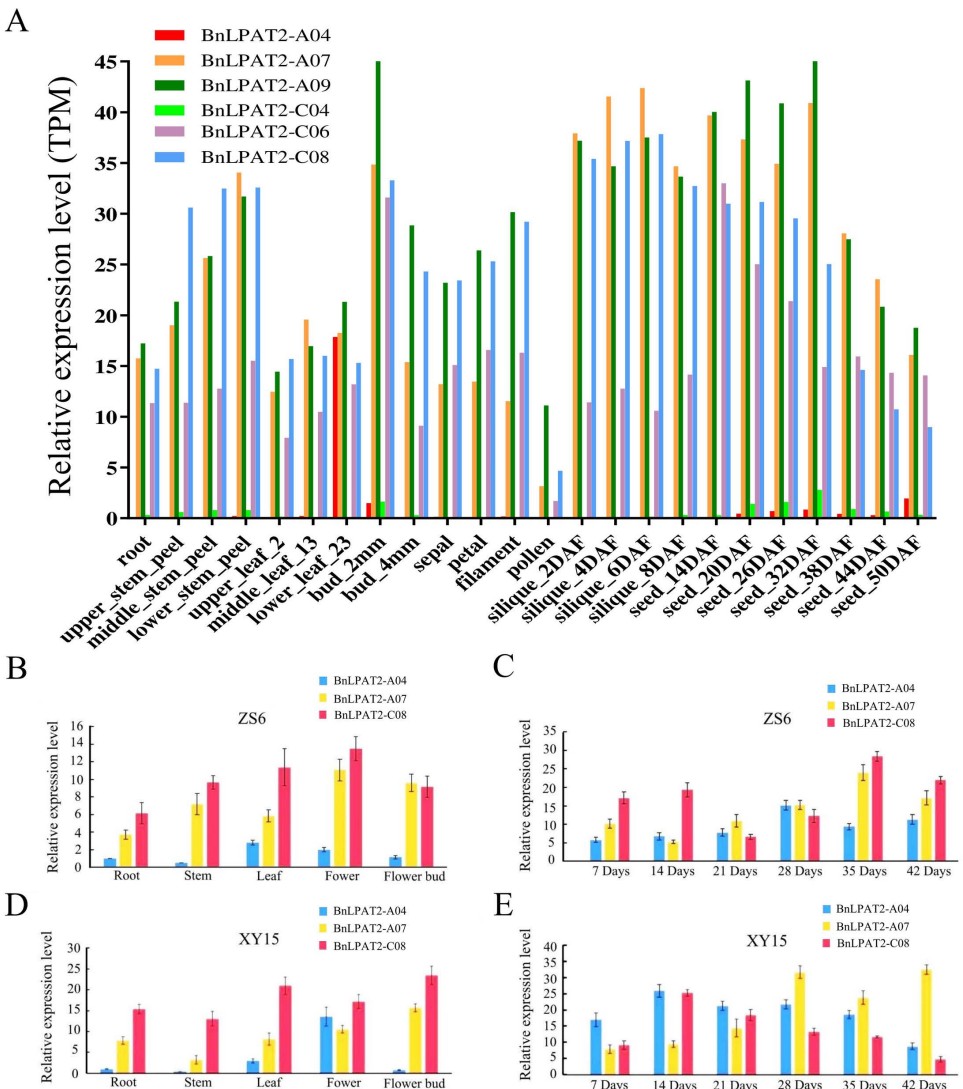

**Fig 4. Expression of *BnLPAT2* genes in different tissues of *B. napus*.** (A) Expression of *BnLPAT2* Genes in ZS11. (B and C) Expression of *BnLPAT2* Genes in ZS6. (D and E) Expression of *BnLPAT2* Genes in XY15.

*BnLPAT2-A04/A07/A09* significantly increased seed oil content compared to the wild type, with *BnLPAT2-A07* transgenic plants showing a 4.8% increase in seed oil content, which was statistically significant (Fig 5A). Fatty acid composition analysis revealed that overexpressing all four *BnLPAT2* genes significantly increased the contents of C18:2 and C18:3 fatty acids. In particular, overexpression of *BnLPAT2-A07* led to a significant increase in Stearic acid (C18:0) fatty acid content, as well as a significant decrease in the contents of Arachidic acid (C20:0) and 9-eicosenoic acid (C20:1) fatty acids (Fig 5B). Thus, *BnLPAT2-A07* had the most pronounced effect on seed oil accumulation and fatty acid composition. Therefore, we selected *BnLPAT2-A07* and constructed a *Napin* promoter driven overexpression vector to transform ZS6.

We found that the oil content in two *CaMV35S* driven overexpression lines and two *Napin* promoter-driven overexpression lines was significantly higher than that of ZS6, specifically BnLPAT2-35S-1, BnLPAT2-35S-2, BnLPAT2-NP-1, and BnLPAT2-NP-3. Among them,

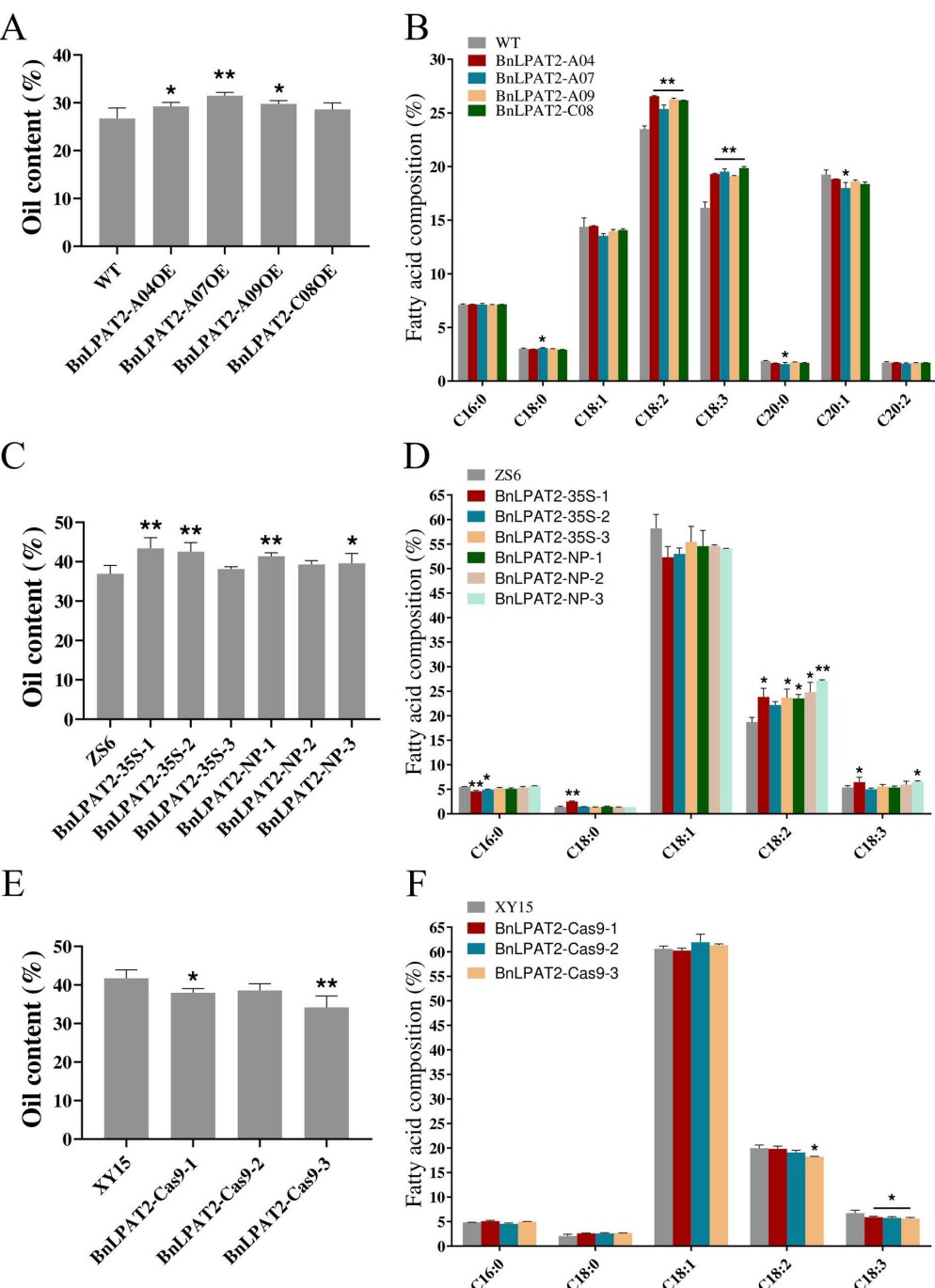

**Fig 5. Detection of oil content and fatty acid content in seeds.** (A and B) Oil content and fatty acid content in *BnLPAT2* gene-transferred Arabidopsis seeds. (C and D) Oil content and fatty acid content in *B. napus* seeds overexpressing the *BnLPAT2* gene. (E and F) Oil content and fatty acid content in *B. napus* seeds with *BnLPAT2* gene knockout. * indicates **p** < 0.05, ** indicates **p** < 0.01.

the seed oil content in the BnLPAT2-35S-1 line was significantly increased by 6.4% compared to the control (Fig 5C). We analyzed the fatty acid content in the transgenic lines' seeds (Fig 5C) and found that the C18:2 content was significantly increased in all lines except BnLPAT2-35S-2. The C18:3 content was significantly increased in BnLPAT2-35S-1 and BnLPAT2-NP-3 lines. The C16:0 content was significantly reduced in BnLPAT2-35S-1 and BnLPAT2-35S-2 lines, while the C18:0 content was significantly increased in the BnLPAT2-35S-1 line. Our results indicate that overexpression of the *BnLPAT2-A07* gene in rapeseed promotes the accumulation of C18:2 and C18:3 fatty acids.

To further validate the effect of the *BnLPAT2-A07* gene on oil accumulation, we used CRISPR/Cas9 technology to knock out four *BnLPAT2* genes in rapeseed (the vector construction is shown in S1 Fig). We assessed the editing efficiency of the four homologous copies of the *BnLPAT2* gene in the $T_0$ generation of CRISPR/Cas9-edited rapeseed. The detection primers for CRISPR/Cas9 editing in rapeseed and the gene editing efficiency detection are shown in S5 and S6 Tables. The proportion of rapeseed plants with edits at the homologous copies ranged from 5.56% to 22.22%, with an average editing efficiency of 16.67%; the editing efficiency at Target 1 ranged from 0% to 22.22%, with an average of 9.72%; the editing efficiency at Target 2 ranged from 5.56% to 16.67%, with an average of 12.50%. Sequencing alignment revealed that the editing types at the four homologous copies of the *BnLPAT2* gene in the $T_0$ generation lines BnLPAT2-Cas9–1 (401-1), BnLPAT2-Cas9–2 (401–2), and BnLPAT2-Cas9–3 (401–3) were more diverse compared to other lines. Therefore, these three knockout lines were selected and grown to the $T_2$ generation for seed collection and oil content and fatty acid analysis. The seed oil content of the three knockout lines was lower than that of the control XY15. Among them, the oil content of BnLPAT2-Cas9–1 was significantly lower than that of the control, and BnLPAT2-Cas9–3 had a 7.5% reduction in oil content, which was statistically significant (Fig 5E). Fatty acid analysis showed that the content of C18:3 in the seeds of all three knockout lines was significantly lower than that of the control, and only the seeds of BnLPAT2-Cas9–3 showed a significant reduction in C18:2 content (Fig 5F). Knocking out the *BnLPAT2* gene not only reduced oil content but also decreased the levels of C18:2 and C18:3.

## Transcriptome sequencing analysis of transgenic *BnLPAT2 B. napus* seeds during development

To investigate the biological function of the *BnLPAT2* gene in promoting seed oil accumulation, we conducted transcriptome sequencing analysis on seeds collected 30 days after flowering from *B. napus* overexpression and knockout lines. We found that the DEGs induced by *BnLPAT2* overexpression driven by the *Napin* promoter were the fewest, while the *CaMV35S* promoter overexpression and knockout lines showed a larger number of DEGs, primarily characterized by upregulated gene expression (Fig 6A). A total of 4,222 DEGs were identified as common among the three *BnLPAT2* knockout lines (Fig 6B). Further Gene Ontology (GO) enrichment analysis revealed that these common DEGs were significantly enriched in biological processes, including "seed maturation" and "fatty acid biosynthetic process" (Fig 6C). In the seeds of both *Napin* and *CaMV35S* promoter-driven overexpression lines, 1,113 DEGs were identified as common (Fig 6D). GO enrichment analysis showed that these DEGs were significantly enriched in 19 biological processes, including "fatty acid biosynthetic process," "seed maturation," "very long-chain fatty acid biosynthetic process," and "very long-chain fatty acid metabolic process" (Fig 6E). We observed that the "fatty acid biosynthetic process" was significantly enriched in both overexpression and knockout lines. Additionally, the overexpression of the *BnLPAT2* gene significantly impacted the "very long-chain fatty acid biosynthetic process" and "very long-chain fatty acid metabolic process," providing a biological explanation for the increased content of C18:2 and C18:3.

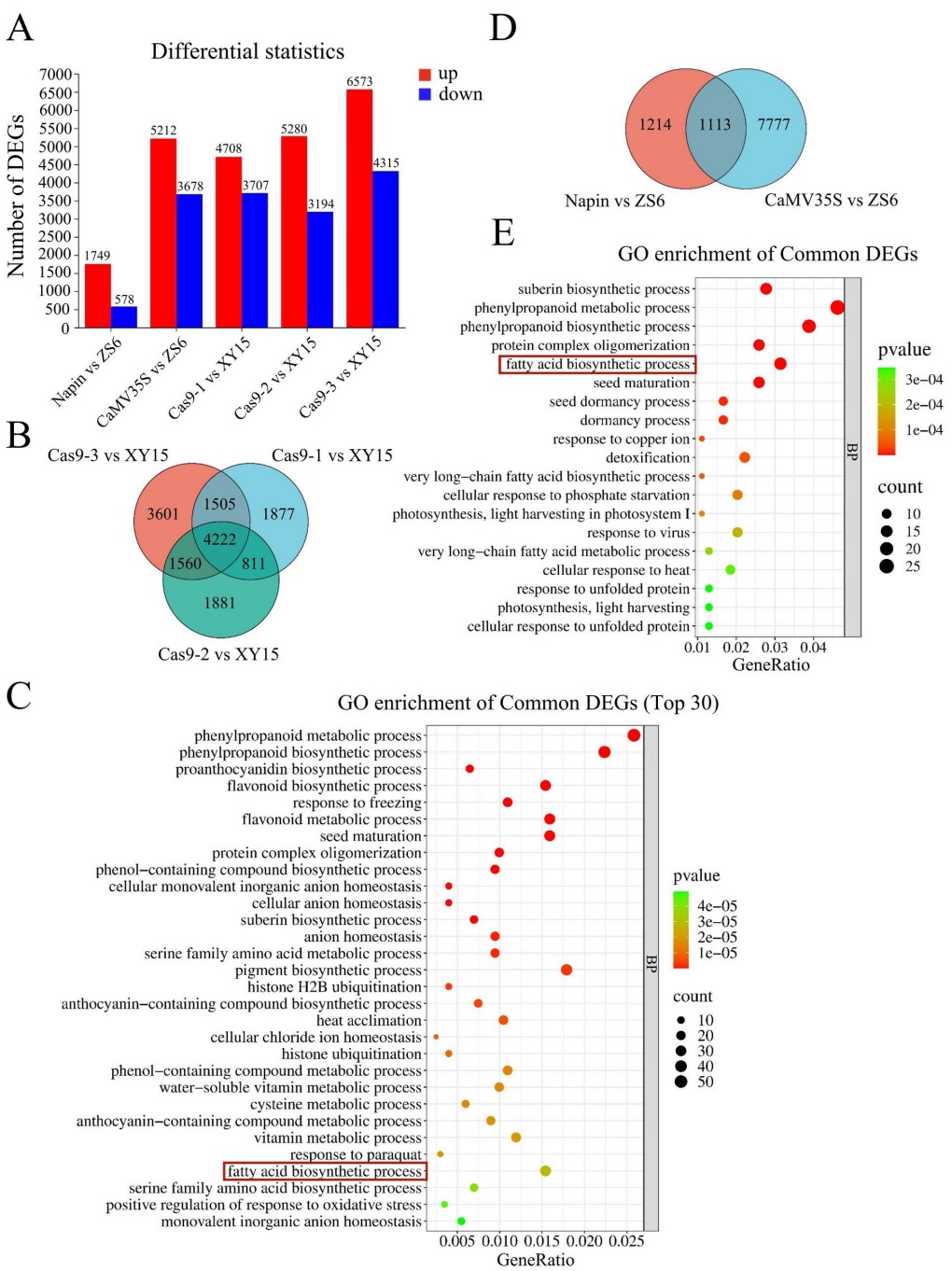

**Fig 6. Analysis of transcriptome data from transgenic rapeseed seeds.** (A) Statistics of differential gene numbers. (B and C) Venn diagram of differential genes in knockout lines and GO enrichment analysis. (D and E) Venn diagram of differential genes in overexpression lines and GO enrichment analysis.

## Identification of key genes interacting with the *BnLPAT2* gene to regulate lipid accumulation

Based on the transcriptome analysis results, we generated an expression heatmap for 44 genes involved in the fatty acid biosynthetic process in overexpression and knockout lines. We identified seven genes that were upregulated in the overexpression lines but downregulated in

the knockout lines. These genes are *BnaC07G0374200ZS* (Cytochrome P450 86B1, *CYP86B1*), *BnaA01G0399900ZS* (Acyl carrier protein, *ACP1*), *BnaA09G0139900ZS* (Long chain acyl-CoA synthetase 3, *LACS3*), *BnaA05G0181300ZS* (Long chain acyl-CoA synthetase 2, *LACS2*), *BnaC06G0093600ZS* (Cytochrome P450 86B1, *CYP86B1*), *BnaA03G0402500ZS* (Cytochrome P450 86B1, *CYP86B1*), and *BnaA02G0390300ZS* (Cytochrome P450 86B1, *CYP86B1*) (Fig 7A). Based on the transcriptome data from both *BnLPAT2* gene overexpression and knock-out lines, we conducted a protein-protein interaction (PPI) network analysis for the selected gene set and constructed PPI networks. Two protein interaction network diagrams were constructed (Fig 7B and C). According to the degree of interaction of these proteins, we focused on hub proteins with more than ten connections. Fig 7B shows that several proteins in the RPS13B (ribosomal protein US15Y) family are core proteins, interacting extensively with RPL24B (ribosomal protein EL24Y), UTP18 (U3 small nucleolar rna-associated protein 18), RPL29B (ribosomal protein EL29Y), RPL35AA (ribosomal protein EL33W), MCM5 (minichromosome maintenance 5), and RPS24A (ribosomal protein ES24Z). Similarly, Fig 7C reveals that several RPS13B family proteins are core proteins, interacting extensively with RPS29A (ribosomal protein US14X), RPS24A, MCM5, RPL29B, and RPL24B. Interestingly, most of these core proteins are ribosomal proteins. The genes encoding RPS13B, RPS24A, RPL24B, and RPL35AA proteins exhibited relatively high expression levels (Fig 7D).

## Discussion

Plant oils are primarily composed of triacylglycerols, representing the most energy-dense form of biological carbon storage [36]. Lysophosphatidic acid acyltransferase (LPAT) is the second acyltransferase in the triacylglycerol biosynthesis pathway, characterized by strong substrate selectivity and specificity, with a marked preference for unsaturated fatty acids [37]. LPAT plays a role in regulating plant oil accumulation and is closely associated with seed oil content [38,39]. In addition, it is involved in optimizing fatty acid composition, the synthesis of membrane lipids and signaling molecules [40], influencing seed embryo development [14], and contributing to plant stress resistance [41,42]. In this study, four *BnLPAT2* genes were cloned from *B. napus*. *BnLPAT2-A04* encodes 384 amino acids, while *BnLPAT2-A07/A09/C08* encode 390 amino acids. The four *BnLPAT2* genes share the same conserved domain and belong to the PLN02380 superfamily. The protein sequences of BnLPAT2-A07/A09/C08 contain 10 motifs, which are consistent in arrangement with those in Arabidopsis AtLPAT2. However, the BnLPAT2-A04 protein sequence lacks one motif. This motif deletion might result in functional differences between BnLPAT2-A04 and the other three homologous copies. Our study revealed that the homologous copies of the *BnLPAT2* gene exhibit distinct tissue expression patterns. *BnLPAT2-A04* is relatively highly expressed in reproductive organs but has extremely low expression levels in vegetative organs. *BnLPAT2-C08/A09* shows high expression levels in various tissues of *B. napus*. Similarly, *BnLPAT2-A07* is expressed at relatively high levels across tissues, with a trend of increased expression during silique and seed development. These findings suggest that the homologous copies of the *BnLPAT2* gene may have organ-specific roles, with their contributions to glycerolipid biosynthesis and plant growth varying across different organs.

According to the results of oil content and fatty acid composition analysis, overexpression of *BnLPAT2-A04/A07/A09/C08* in *Arabidopsis thaliana* increased seed oil content by 2.53%, 4.78%, 3.05%, and 1.91%, respectively. Although overexpression of all four *BnLPAT2* homologs increased oil content, the increase observed for *BnLPAT2-C08* did not reach statistical significance, and the extent of oil content enhancement varied among the copies. Only over-expression of *BnLPAT2-A07* significantly and substantially increased seed oil content. These

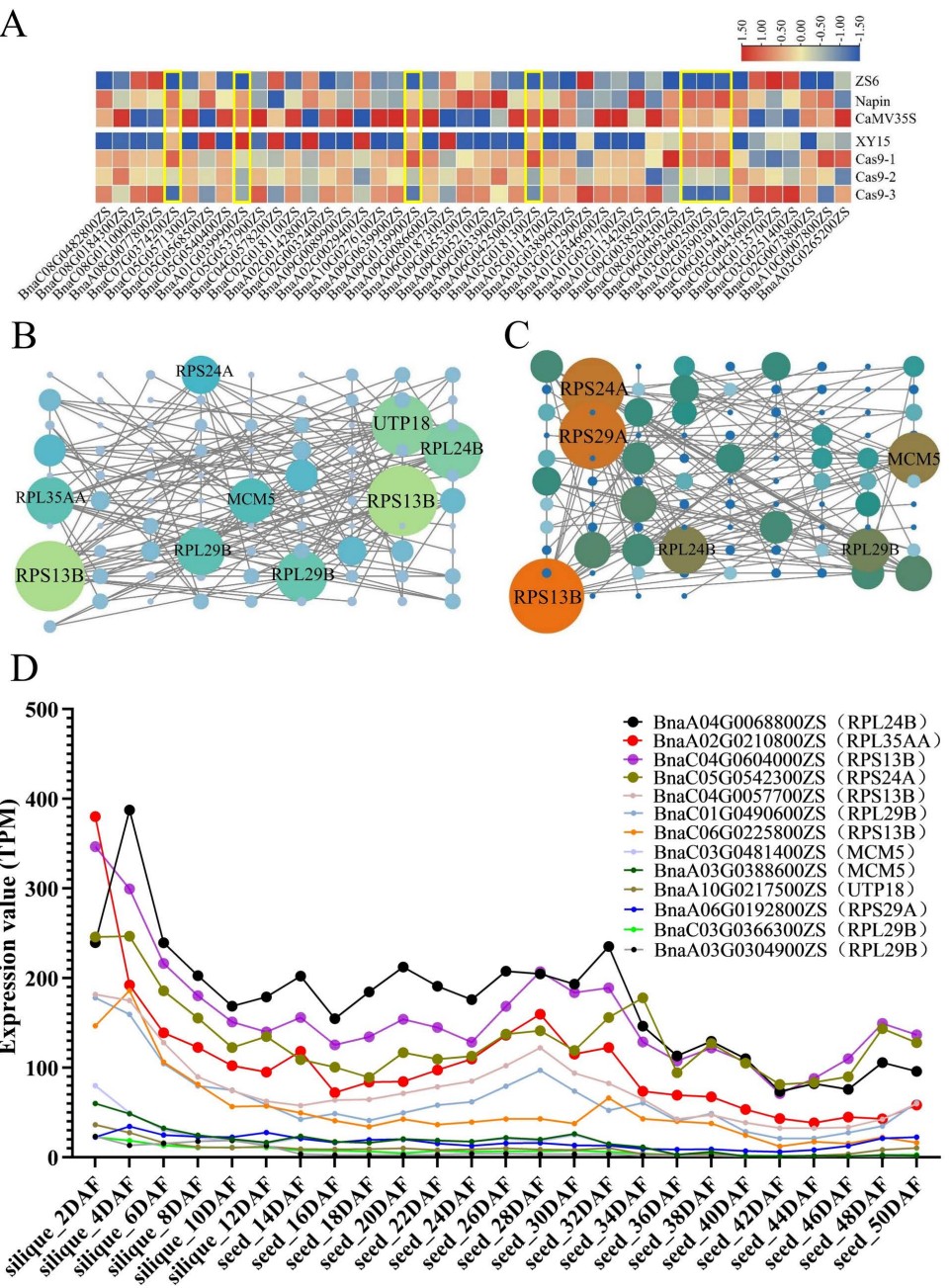

**Fig 7. Mining transcriptome data to identify key genes regulated by *BnLPAT2* in seed oil accumulation.** (A) Heatmap of DEGs expression in the fatty acid biosynthetic process. (B) Protein interaction network analysis results of the differential gene set in overexpressed transgenic rapeseed seeds. (C) Protein interaction network analysis results of the differential gene set in rapeseed knockout line seeds. Note: Nodes represent proteins, and edges represent interactions between proteins. The size of a node is proportional to its connectivity; the more edges connected to a node, the higher its degree and the larger the node. This indicates that the core proteins play a more significant role in the network. (D) The expression of genes encoding core proteins in siliques and seeds of ZS11.

findings indicate functional differences among the four *BnLPAT2* genes in seed oil accumulation, but they share a common feature: all significantly increased the levels of C18:2 and C18:3 fatty acids, thereby altering the fatty acid composition. This result aligns with studies on heterologous expression of *LPAT*. Chen [30] cloned and studied the expression characteristics of the peanut *AhLPAT2* gene, finding a strong correlation between *AhLPAT2* transcription levels and seed oil content. Specific overexpression of *AhLPAT2* in *Arabidopsis* seeds significantly increased the total oil yield and seed weight of transgenic offspring. Overexpression of *RcLPAT2* in castor beans increased the proportion of 18:1OH at the sn-2 position of triacylglycerols from 2% to 14–17%, which led to an increase in HFA-TAGs (hydroxy fatty acid- triacylglycerol) from 5% to 13–14% [31].

We propose that *BnLPAT2-A07* is the primary functional homolog contributing to seed oil accumulation in *B. napus*. By generating *BnLPAT2-A07* overexpressing transgenic lines, the seed oil content was increased by 4.46% to 6.44%. Specific expression of *BnLPAT2-A07* in seeds elevated C18:2 levels by up to 8.4% and C18:3 levels by 1.11%. Using CRISPR-Cas9 technology, we further validated the functional role of *BnLPAT2-A07* in regulating fatty acid and oil accumulation in *B. napus* seeds, which is consistent with the findings of Zhang [27]. Previous studies have shown that LPAT exhibits strong substrate specificity during triacylglycerol assembly, with a marked preference for unsaturated fatty acyl-CoAs such as C18. Castor bean *RcLPAT2* [20] and *Arabidopsis AtLPAT2* prefer 18:1-CoA [5]. Our results demonstrate that overexpression or knockout of *BnLPAT2-A07* in *B. napus* significantly alters C18:3 levels in seeds, suggesting that *BnLPAT2* has a preference for C18:3 acyl substrates. This conclusion aligns with Chen [43], who hypothesized that *BnLPAT2* has a preference for linolenic acid. However, in *BnLPAT2-OE Arabidopsis* lines, Chen also observed an increase in long-chain fatty acids (C20 and above), suggesting that *BnLPAT2* may also have a preference for long-chain fatty acids.

*LPAT2* plays a crucial role in *Arabidopsis thaliana*, where *AtLPAT2* is essential for glycerolipid biosynthesis, growth, and development, with knockout mutants exhibiting lethal phenotypes. The knockout mutants of *AtLPAT2* display lethal phenotypes [14,21]. It functions in the Kennedy pathway in the endoplasmic reticulum, showing broad tissue expression across developmental stages [21]. In *B. napus*, transcriptome analysis of 30-day-old seeds from *BnLPAT2* transgenic lines revealed numerous DEGs enriched in pathways related to fatty acid biosynthesis, including "fatty acid biosynthetic process" and "very long-chain fatty acid biosynthetic process." Seven key genes potentially linked to oil biosynthesis were identified, including *CYP86B1*, *ACP1*, and *LACS2/3*. *CYP86B1* is a hydroxylase for very long-chain fatty acids and has been demonstrated to be involved in the biosynthesis of ultra-long-chain fatty acids in plants [44]. Upregulated expression of *ACP1* has also been associated with increased seed oil content, as evidenced by research on the overexpression of peanut *AhLPAT2* in *Arabidopsis* [30]. Long-chain acyl-CoA synthetases (*LACS*) are key enzymes involved in fatty acid metabolism in plants, including phospholipid biosynthesis and TAG biosynthesis [45,46]. *BnLACS2* has been confirmed to participate in oil biosynthesis in *B. napus* seeds, where its overexpression increases seed oil content, while *BnLACS2-RNAi* lines show reduced seed oil content [47]. These findings suggest that *BnLPAT2* enhances oil content by promoting long-chain fatty acid synthesis, such as C18:2 and C18:3.

Through protein interaction network analysis, we identify some core proteins in the BnLPAT2 transgenic lines, most of which are ribosomal proteins. These proteins are likely to be important in fatty acid or lipid synthesis. Ribosomal proteins not only participate in protein synthesis but also play key roles in lipid metabolism. They affect lipid synthesis and degradation by regulating the expression of metabolism-related genes or interacting with signaling pathways. Ribosomal proteins regulate the expression of lipid

metabolism-related genes by influencing transcription factors or signaling pathways (such as mechanistic target of rapamycin, mTOR and sterol regulatory element-binding protein, SREBP) [48,49]. For example, Ribosomal protein S6 kinase (S6K) regulates the expression of fatty acid synthase (FAS) and acetyl-CoA carboxylase (ACC) through the mTOR signaling pathway, promoting de novo lipid synthesis [50]. In plants, ribosomal proteins affect the synthesis and accumulation of fatty acids and triglycerides by regulating the expression of lipid metabolism-related genes. Studies show that mutations in ribosomal large subunit 4 (*rpl4d*) lead to downregulation of lipid metabolism genes, and lipid accumulation decreases in the meristem of *rpl4d* mutants [51]. Target of rapamycin (TOR) signaling pathway plays a central role in plant cell growth and metabolism, with Ribosomal protein S6 kinase (S6K) regulating the synthesis of galactolipids in rice through this pathway [52]. Additionally, ribosomal synthesis and post-translationally modified peptides can form hybrid lipids with fatty acids, thereby participating in fatty acid biosynthesis pathways [53,54]. As can be seen, the involvement of ribosomal proteins in lipid synthesis is a highly complex biological process.

## Conclusions

This study cloned four *BnLPAT2* genes (*BnLPAT2-A04/A07/A09/C08*) from *B. napus*. BnLPAT2-A07/A09/C08 share highly conserved motifs with the Arabidopsis AtLPAT2 protein. All four *BnLPAT2* genes influence seed oil content, with *BnLPAT2-A07* showing the most significant impact on seed oil accumulation and fatty acid content, being highly expressed during the late stages of seed development. Transcriptome analysis identified considerable enrichment in biological processes such as the "fatty acid biosynthetic process", "very long-chain fatty acid biosynthetic process", and "very long-chain fatty acid metabolic process". We find some ribosomal proteins that may be involved in lipid synthesis through protein interaction network analysis. Our results suggest that *BnLPAT2* genes may enhance seed oil content by promoting the production of long-chain fatty acids like C18:2 and C18:3. This finding is significance for improving the fatty acid composition and oil content of *B. napus* seeds. Therefore, our study provides valuable insights for genetic engineering to modify oilseed crop oil composition and increase seed oil content.

## Supporting information

**S1 Table.  Primers required for *BnLPAT2* gene cloning.**
(XLSX)

**S2 Table.  Overexpressed *BnLPAT2* and CRISPR/ Cas9 primers were identified and detected.**
(XLSX)

**S3 Table.  Sequences of primers for *BnLPAT2* expression.**
(XLSX)

**S4 Table.  Physicochemical properties analysis of four *BnLPAT2* in *Brassica napus*.**
(XLSX)

**S5 Table.  Prediction protein secondary structure of BnLPAT2 in *Brassica napus*.**
(XLSX)

**S6 Table.  Detection of the efficiency of editing *BnLPAT2* homologous gene and target site using CRISPR/ CAS9 of $T_0$ generation in rapeseed.**
(XLSX)

**S7 Table. The transcriptome sequencing annotation of all genes and their TPM expression levels.**
(XLS)

**S1 Fig. Construction and detection of expression vector.**
(DOCX)

**S1 Raw Images. The original and uncropped images supporting all blot and gel results reported in an article's figures and supporting information files.**
(PDF)

## Author contributions

**Conceptualization:** Luyao Huang, Man Xing, Xinghua Xiong.

**Data curation:** Luyao Huang.

**Formal analysis:** Luyao Huang, Zhiqiang Liao.

**Funding acquisition:** Luyao Huang, Man Xing, Xinghua Xiong.

**Investigation:** Yujing Zou, Yong Liu, Huihui Wang.

**Methodology:** Zhiqiang Liao, Man Xing, Xinghua Xiong.

**Project administration:** Man Xing, Xinghua Xiong.

**Resources:** Xinghua Xiong.

**Software:** Luyao Huang, Leping Zou, Sun Liang, Yu Kang.

**Supervision:** Luyao Huang, Xinghua Xiong.

**Validation:** Yu Kang, Tuo Chen, Man Xing.

**Visualization:** Luyao Huang, Sun Liang, Shan Tong.

**Writing – original draft:** Luyao Huang, Yong Liu, Man Xing.

**Writing – review & editing:** Man Xing, Xinghua Xiong.

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
