## [Decision Letter · Decision Letter 0]

27 Jan 2025

PONE-D-24-59454BnLPAT2 Gene Regulates Oil Accumulation in Brassica napus

by Modulating Linoleic and Linolenic Acid Levels in SeedsPLOS ONE

Dear Dr. Xing,

Thank you for submitting your manuscript to PLOS ONE. After careful consideration, we feel that it has merit but does not fully meet PLOS ONE’s publication criteria as it currently stands. Therefore, we invite you to submit a revised version of the manuscript that addresses the points raised during the review process.

We look forward to receiving your revised manuscript.

Kind regards,

Bahram Heidari

Academic Editor

PLOS ONE

Journal requirements: When submitting your revision, we need you to address these additional requirements. 1. Please ensure that your manuscript meets PLOS ONE's style requirements, including those for file naming. The PLOS ONE style templates can be found at https://journals.plos.org/plosone/s/file?id=wjVg/PLOSOne_formatting_sample_main_body.pdf and https://journals.plos.org/plosone/s/file?id=ba62/PLOSOne_formatting_sample_title_authors_affiliations.pdf. 2. PLOS ONE now requires that authors provide the original uncropped and unadjusted images underlying all blot or gel results reported in a submission’s figures or Supporting Information files. This policy and the journal’s other requirements for blot/gel reporting and figure preparation are described in detail at https://journals.plos.org/plosone/s/figures#loc-blot-and-gel-reporting-requirements and https://journals.plos.org/plosone/s/figures#loc-preparing-figures-from-image-files. When you submit your revised manuscript, please ensure that your figures adhere fully to these guidelines and provide the original underlying images for all blot or gel data reported in your submission. See the following link for instructions on providing the original image data: https://journals.plos.org/plosone/s/figures#loc-original-images-for-blots-and-gels.   In your cover letter, please note whether your blot/gel image data are in Supporting Information or posted at a public data repository, provide the repository URL if relevant, and provide specific details as to which raw blot/gel images, if any, are not available. Email us at plosone@plos.org if you have any questions. 3. PLOS requires an ORCID iD for the corresponding author in Editorial Manager on papers submitted after December 6th, 2016. Please ensure that you have an ORCID iD and that it is validated in Editorial Manager. To do this, go to ‘Update my Information’ (in the upper left-hand corner of the main menu), and click on the Fetch/Validate link next to the ORCID field. This will take you to the ORCID site and allow you to create a new iD or authenticate a pre-existing iD in Editorial Manager. 4. We note that the grant information you provided in the ‘Funding Information’ and ‘Financial Disclosure’ sections do not match.  When you resubmit, please ensure that you provide the correct grant numbers for the awards you received for your study in the ‘Funding Information’ section. 5. Thank you for stating the following financial disclosure:  [This work was supported by the Natural Science Foundation of China (32401904), the Natural Science Foundation of Chongqing (CSTB2024NSCQ-MSX0906), Special fund for guiding city and county science and technology development of Jiangxi Province, and the National Key Research and Development Program of China (2017YFD0101700). ].  Please state what role the funders took in the study.  If the funders had no role, please state: ""The funders had no role in study design, data collection and analysis, decision to publish, or preparation of the manuscript."" If this statement is not correct you must amend it as needed. Please include this amended Role of Funder statement in your cover letter; we will change the online submission form on your behalf.

Reviewers' comments:

Reviewer's Responses to Questions

**Comments to the Author**

1. Is the manuscript technically sound, and do the data support the conclusions?

Reviewer #1: Yes

Reviewer #2: Yes

2. Has the statistical analysis been performed appropriately and rigorously? 

Reviewer #1: Yes

Reviewer #2: Yes

3. Have the authors made all data underlying the findings in their manuscript fully available?

Reviewer #1: Yes

Reviewer #2: Yes

4. Is the manuscript presented in an intelligible fashion and written in standard English?

Reviewer #1: Yes

Reviewer #2: Yes

5. Review Comments to the Author

Reviewer #1: Manuscript Number: PONE-D-24-59454

Manuscript Title: BnLPAT2 Gene Regulates Oil Accumulation in Brassica napus by Modulating Linoleic and Linolenic Acid Levels in Seeds

Reviewer:

Dear editor,

I thoroughly reviewed the manuscript titled "BnLPAT2 Gene Regulates Oil Accumulation in Brassica napus by Modulating Linoleic and Linolenic Acid Levels in Seeds." This paper explores the functional analysis of BnLPAT2 genes using bioinformatics, qRT-PCR, CRISPR/Cas9, overexpression, and transcriptome sequencing. The findings suggest that these genes play a crucial role in regulating plant oil accumulation, optimizing fatty acid composition, and contributing to the synthesis of membrane lipids and signaling molecules. While the paper is well-executed and suitable for publication in PONE, I recommend major revisions before publication to address some necessary corrections and edits. One major concern is the lack of detail in the materials and methods section. I suggest that the authors provide more information about the varieties used, transgenic lines, experimental conditions, and plant growth condition to ensure the reproducibility of the study. Overall, the study presents valuable insights into the role of BnLPAT2 genes in oil accumulation in Brassica napus. With some revisions and additions to the materials and methods section, the paper will be better positioned for publication.

Abstract:

- Please include the full names of qRT-PCR and CRISPR/Cas9 in the abstract.

- When selecting keywords, it is important to choose words that are distinct from the title. It is recommended to use a variety of keywords to enhance search engine optimization.

- In the following sentences, 'Overexpression of these genes increased seed oil content and the proportion of C18:2/C18:3 fatty acids, with BnLPAT2-A07 achieving a 6.4% increase in oil content,' the seed oil enhancement should be revised to reflect an increase in seed oil content ranging from 4.46% to 6.44%. This adjustment aligns with the results and discussion section of the study.

Introduction:

- In the following sentences, Zhang utilized CRISPR/Cas9 gene-editing technology to conduct targeted editing of multiple homologous copies of BnLPAT2 and BnLPAT5 in rapeseed [27]. Similarly, Chen introduced the peanut AhLPAT2 gene into Arabidopsis, leading to enhancements in seed weight, oil content, total fatty acids, and unsaturated fatty acids [30]. References should be included after the respective names Zhang [27] and Chen [30].

- In the final paragraph of the introduction, the author should address the gap in previous studies regarding LPAT2 expression and seed oil accumulation that prompted the current study. The structure of the study's objectives should include a discussion of the issues identified in previous research, the hypothesis being tested in the present study, and the framework of the current investigation. For example, it was hypothesized that BnLPAT2 may exhibit a preference for linolenic acid and long-chain fatty acids.

M&M

How were Transgenic rapeseed plants produced? It is important to clarify the conditions in which the Transgenic rapeseed plants were cultured. Additionally, more details about the genotypes used, such as XY15, ZS6, and ZS11, need to be provided in a cohesive manner throughout the manuscript. It is also essential to include information about the wild type, transgenic lines (T0, T1, and T2), and the conditions under which they were produced. Knockout lines were grown under specific conditions, and the T2 generation was produced for seed collection, oil content analysis, and fatty acid analysis. However, this information was lacking in the Materials and Methods section. It is crucial for authors to thoroughly present all materials, experimental conditions, and procedures so that other researchers can replicate the study with ease and reliability. This will ensure that the protocol can be followed accurately, leading to consistent and reproducible results.

- The determination of seed oil content should be organized into a cohesive paragraph. Additionally, relevant references for Gas Chromatography should be included to support the analysis.

- Please revise the formula for calculating seed oil content (%) as follows:

[(mass of internal standard FA (C17:0) / proportion of internal standard FA (C17:0)] - (mass of internal standard FA (C17:0) / seed mass)

- The section on RNA extraction should include more details, such as the specific tissue part at which phenological stage (time) the extraction is performed, and from which rapeseed varieties (XY15 and ZS6).

- The author collected siliques at various time points, including 7, 14, 21, 28, 35, and 42 days after pollination. However, in the transcriptome analysis, siliques were specifically collected at 30 days after pollination. It is important to understand the rationale behind these different time points and why only 30 days was chosen for the transcriptome analysis. Please provide clarification on this matter.

- Please include the full names of The Gene Ontology (GO) and Kyoto Encyclopedia of Genes and Genomes (KEGG) databases when first mentioned.

Results

- Please move these sentences to M&M. Total RNA was extracted from XY15 seeds two weeks after pollination (Figure 1 A). Using reverse transcribed cDNA as a template, PCR amplification produced four target bands, as shown in Figure 1 B. The target fragments were recovered, ligated into the pUCm-T vector, and transformed into DH5α (Escherichia coli). After sequencing and alignment, four full-length CDS sequences were cloned and named BnLPAT2-A04 (1155 bp), BnLPAT2-A07 (1173 bp), BnLPAT2-A09 (1173 bp), and BnLPAT2-C08 (1173 bp), encoding proteins of 384, 390, 390, and 390 amino acids, respectively.

- In this sentence “Secondary structure predictions indicated that BnLPAT2 proteins consisted of random coils”, the phrase "consisted of" should be changed to "consist of".

- In this sentence, it is recommended to include the names of the genes associated with α-helices. The α-helices being the most prevalent, representing 46.15% (BnLPAT2-A04), 47.68% (BnLPAT2-A07), 48.21% (BnLPAT2-A09), and 45.57% (BnLPAT2-C08).

- In the following sentence, 'In contrast, BnLPAT2-A07/A09/C08 share 91.82% and 98.04% similarity with AtLPAT2, respectively (Figure 2 B)', please include the similarity of BnLPAT2-A09 with AtLPAT2. Additionally, what is the level of similarity between the three genes?

- This information 'To analyze the potential functions of the four BnLPAT2 genes in rapeseed, the PlantCARE software was used to analyze the promoter sequences in the 2 kb upstream region of the BnLPAT2 genes' should be include in the Materials and Methods section, as it pertains to the methodology used in the study, rather than the results obtained.

- The number of cis-acting elements in the promoter sequences of the BnLPAT2 genes, categorized in each functional group, should be added.

- Please move the following sentences from the results section to the Materials and Methods section:

“Based on the transcriptome data reported in the BnTIR database (https://yanglab.hzau.edu.cn/BnTIR), the expression levels of six homologous copies of the BnLPAT2 gene across various tissues were analyzed using heatmap analysis (Figure 4 A)”

“Total RNA was extracted from roots, stems, leaves, flowers, and seeds of ZS6 and XY15, and cDNA was synthesized by reverse transcription for qRT-PCR analysis”

- During the RNA extraction process, one location focused on extracting RNA from flower buds, while another location only mentioned flowers. Similarly, one location considered extracting RNA from seeds, while another location referred to different developmental stages of seeds. It is important to ensure consistency and coordination throughout the entire manuscript. Please make the necessary corrections to address these discrepancies.

- The study focused on the expression patterns of BnLPAT2 genes in various tissues of B. napus, specifically examining three genotypes: ZS11, ZS6, and XY15. However, the Materials and Methods section did not mention ZS11 at all. It is important for the author to address this oversight and ensure that all relevant information is included throughout the manuscript for coherence and accuracy.

- Please move the sentence "Seeds from the T2 generation of Arabidopsis were harvested to measure oil content and fatty acid composition" to the Materials and Methods section. Additionally, please provide more detailed information about the T2 generation, including how it was produced and under what conditions.

- Why was the average editing efficiency so low, at less than 17%? Could it be possible that the primer used was not suitable, or that the target site was not highly editable by BnLPAT2-Cas9?

- In the transcriptome analysis section, we initially refer to differentially expressed genes (DEG) and gene ontology (GO) by their full names. Subsequently, we use the abbreviations for DEG and GO. It is important to maintain consistency in terminology throughout the manuscript

- Please provide the full names of hub proteins, including RPS13B, RPL24B, UTP18, RPL29B, RPL35AA, MCM5, and RPS24A. For example, ribosomal protein S13B (RPS13B)...

- These sentences “We searched for the genes encoding these proteins in the BnTIR database and examined their expression in the seeds of B. napus ZS11” should be relocated to the Materials and Methods section.

- These findings “Recent studies have reported that ribosome synthesis and peptide post-translational modification can form hybrid lipid peptides with fatty acids, playing crucial roles in the fatty acid biosynthetic pathway [36,37]. Although these genes encoding core proteins show high expression in seeds, their involvement in oil synthesis and accumulation in B. napus seeds requires further investigation.” should be discussed in the subsequent section.

Discussion

- "In line 14, 'AtLPAT2' should be italicized. Please review the manuscript to ensure that gene names are correctly formatted in italics."

- The cis-acting element in promoter analysis is related to oil content. I suggest the authors discuss this element further, as well as delve into the functional analysis focused on the oil regulatory process.

- Please provide the complete name of "tri-HFA-TAGs"

- The author conducted a study to measure the expression level of BnLPAT2 during various stages of seed development in siliques (7, 14, 21, 28, 35, and 42 days after pollination). The results showed that the expression of BnLPAT2 varied at different stages, with higher levels of expression detected only in the mid-to-late stages of seed development, specifically in BnLPAT2-A07. It is important to discuss the differences in expression levels and how they are related to oil accumulation. By understanding the relationship between expression levels of BnLPAT2 and oil accumulation, we can gain insights into the mechanisms underlying seed development and potentially improve oil production in plants.

- Please include a paragraph discussing the PPI network and hub proteins that play a causal role in oil and fatty acid biosynthesis. It is suggested that we thoroughly analyze each part of the results and its relationship to BnLPAT2 and oil biosynthesis.

Conclusion:

In conclusion, rather than reiterating the results, it is recommended to focus on key points such as the phenological stage at which BnLPAT2 exhibited higher expression, hub proteins and DEGs associated with BnLPAT2 and fatty acid biosynthesis, and how BnLPAT2 enhances oil accumulation. These aspects should be highlighted in the conclusion for a more comprehensive understanding of the study.

References:

The genus and species names, as well as the gene names, in the majority of the references are not properly italicized. please double-check references.

Reviewer #2: This manuscript explores the regulatory role of the BnLPAT2 gene in oil content and fatty acid composition in Brassica napus seeds. The study encompasses gene cloning, bioinformatics analysis, gene overexpression and knockout experiments, and transcriptome analysis. The manuscript is comprehensive, with detailed experimental methods and substantial data support, providing theoretical guidance for improving lipid metabolism and fatty acid composition in B. napus. The experiments were well-conducted and represent an important exploration of the functional study of LPAT2 genes. However, I have several comments and suggestions for the authors to consider:

1. Figure 4 shows six copies of the BnLPAT2 gene in Brassica napus. Could the authors clarify why only four of these copies were selected for the cloning study? Providing a rationale would strengthen the study’s context.

2. The LPAT2 gene is crucial for triacylglycerol (TAG) synthesis. The study by Zhang et al. (2022), titled "Lysophosphatidic Acid Acyltransferase 2 and 5 Commonly, but Differently, Promote Seed Oil Accumulation in Brassica napus," also investigated the BnLPAT2 gene. Could the authors elaborate on the similarities and differences between their work and the findings of Zhang et al.?

3. Considering that RNA-Seq was performed, some essential sequencing statistics, such as QC20 and QC30 scores, were not provided. I recommend that the authors include these data in the manuscript or upload the sequencing data to a public repository to ensure transparency.

4. I suggest a thorough review of manuscript details. For example, in the section “Cloning and Bioinformatics Analysis of Four BnLPAT2 Genes in B. napus,” a hyperlink to the TAIR database should be provided. Additionally, in the “RNA Extraction and qRT-PCR Analysis” section, gene names such as Actin2.1 should be italicized to maintain scientific accuracy.

5. The results section contains some repetitive descriptions, which impacts readability. I suggest restructuring this section to present the findings more concisely and effectively.

6. Certain figures lack statistical significance annotations (e.g., p-values), which affects the credibility of the results. I recommend adding these annotations to enhance data reliability.

7. There are some grammatical errors and overly complex sentences that impact readability. A language review to simplify complex sentences and correct errors would improve the manuscript’s overall clarity and flow.

Overall, this manuscript presents valuable insights into the functional study of the BnLPAT2 gene in oil accumulation and fatty acid metabolism in Brassica napus. Addressing the points mentioned above would significantly enhance the manuscript’s clarity, rigor, and impact.

6. PLOS authors have the option to publish the peer review history of their article (what does this mean? ). If published, this will include your full peer review and any attached files.

**Do you want your identity to be public for this peer review?** For information about this choice, including consent withdrawal, please see our Privacy Policy .

Reviewer #1: **Yes: ** Maryam Salami

Reviewer #2: No

---

## [Author Response · Author response to Decision Letter 0]

22 Feb 2025

Dear Reviewer #1,

Thank you very much for your feedback! Your critiques and suggestions have been very helpful to our research. We can see that you have reviewed our manuscript very carefully, which shows that you paid great attention to our work. We greatly appreciate you taking the time to provide us with these valuable suggestions, which will benefit our future research and paper writing. We will carefully revise the manuscript based on your recommendations. Once again, thank you！

Response Abstract: Based on your suggestions, we have added the full names of qRT-PCR and CRISPR/Cas9, included additional keywords, and revised some of the wording in the abstract. We have adjusted the increase in seed oil content to 4.46% to 6.44%, making it consistent with the results and discussion sections.

Response Introduction: We have carefully revised the citation format of the references and added content to the last paragraph of the Introduction as per your suggestion.

Response Abstract: Based on your suggestions, we have added the full names of qRT-PCR and CRISPR/Cas9, included additional keywords, and revised some of the wording in the abstract. We have adjusted the increase in seed oil content to 4.46% to 6.44%, making it consistent with the results and discussion sections.

Response Introduction: We have carefully revised the citation format of the references and added content to the last paragraph of the Introduction as per your suggestion.

Response M&M:

-Our transgenic Arabidopsis and rapeseed were generated by transformation and selection methods referenced in [33][34], so we did not provide detailed methods in this manuscript. Based on your suggestion, we have included more detailed information about the production and cultivation of the transgenic plants. The obtained T0 generation and the selected T1 and T2 plants were all grown in a plant growth chamber at 24°C with a light cycle of 16 h/8 h. The detailed information has been added to the M&M section.

-We have added the method steps for fatty acid and oil content determination by gas chromatography used in this experiment. The details of RNA extraction have also been thoroughly supplemented. We chose seeds from 30 days after pollination for transcriptome data because the period of rapid lipid accumulation in Brassica napus seeds occurs between 25 and 35 days. Through qRT-PCR, we found that the expression of the BnLPAT2 gene in seeds at 7, 14, 21, 28, 35, and 42 days follows a dynamic change process. Therefore, we selected seeds at 30 days, during the period of rapid oil accumulation, for transcriptome sequencing analysis.

-We have included the full names of terms such as Gene Ontology (GO) and Kyoto Encyclopedia of Genes and Genomes (KEGG) databases the first time they are mentioned.

Response Results:

-Based on your suggestions, we have added the names of genes related to α-helix. Regarding your comment about some sentences in the Results section that should be moved to the M&M section, we have carefully revised this and transferred certain sentences from the Results to the M&M section. You can refer to our revised manuscript for specific changes.

-We have clarified in the manuscript that since the amino acid sequences of BnLPAT2-A09 and BnLPAT2-C08 are identical, the sentence "In contrast, BnLPAT2-A07/A09/C08 share 91.82% and 98.04% similarity with AtLPAT2, respectively (Figure 2 B)" already includes the similarity between BnLPAT2-A09 and AtLPAT2. We believe that the comparison of amino acid sequence differences is more important, so we did not display the gene comparison. Additionally, to some extent, amino acid sequence differences reflect gene sequence differences.

-Regarding your comment about the absence of ZS11 in the Materials and Methods section, it is because we did not grow the ZS11 rapeseed; instead, we used gene expression data of ZS11 downloaded from the BnIR database for comparative analysis.

-Our gene editing efficiency in the T0 generation is around 20%. We do not consider this gene editing efficiency to be too low. Of course, our technique might not be perfect, but nevertheless, we successfully obtained the edited plants with the target gene.

-We have added the full names of the proteins RPS13B, RPL24B, UTP18, RPL29B, RPP35AA, MCM5, and RPS24A. Additionally, we have moved the following sentence "Recent studies have reported that ribosome synthesis and peptide post-translational modification can form hybrid lipid peptides with fatty acids, playing crucial roles in the fatty acid biosynthetic pathway [36,37]. Although these genes encoding core proteins show high expression in seeds, their involvement in oil synthesis and accumulation in B. napus seeds requires further investigation." to the Discussion section.

Response Discussion:

-In line 14, "AtLPAT2" refers to the protein, so it is not italicized. We have carefully reviewed the gene names throughout the manuscript and made corrections where necessary.

-Based on your suggestions, we have revised the Discussion, particularly regarding the PPI network and core proteins. We have added a section discussing the role of ribosomal proteins in fatty acid and oil synthesis. Please review this addition in the revised manuscript.

Response References:

-We have properly italicized the species names and gene names in the references.

Dear Reviewer #2,

Thank you very much for your recognition of our research work. Your criticisms and suggestions will further help in the development of our future research. Below are our responses to your comments and suggestions on this study:

Response 1: Brassica napus is an allopolyploid species, which typically has multiple gene copies. Our initial research involved searching for the AtLPAT2 (AT3G57650) in the TAIR database, then transferring the Arabidopsis gene IDs to the Brassica rapa and Brassica oleracea databases. We used homologous alignment to identify the BnLPAT2 genes in Brassica napus. At that time, very few Brassica napus genome sequences were published, and ultimately, we were able to successfully clone and identify four BnLPAT2 genes.

Response 2：That study investigates the role of Lysophosphatidic Acid Acyltransferase (LPAT) enzymes, BnLPAT2 and BnLPAT5, in seed oil accumulation (SOC) in Brassica napus. The research demonstrates that overexpressing BnLPAT2 and BnLPAT5 significantly increases SOC, while knocking down or knocking out these genes results in a decrease in SOC. BnLPAT2 enhances diacylglycerol synthesis to increase SOC, while BnLPAT5 primarily promotes phosphatidic acid (PA) synthesis for membrane lipids. The transcriptome analysis in this article also highlights DEGs related to SOC involved in processes such as fatty acid and long-chain fatty acid biosynthesis. However, the article does not present results or discussions regarding the regulation of fatty acid synthesis by the BnLPAT2 genes. A large number of reports have shown that the introduction of LPAT2 into Arabidopsis not only increases seed oil content but also significantly alters the levels of linoleic acid, alpha-linolenic acid, and other fatty acids, although the underlying mechanisms remain unresolved. Therefore, in our article, we present the role of BnLPAT2 in seed oil synthesis and fatty acid accumulation and provide a detailed discussion of this process.

Response 3: With the development of technology, transcriptome sequencing has become very common. In our article, we presented important transcriptome analysis results, and we are preparing to upload the transcriptome sequencing data to the NCBI database. If needed, you can contact the authors via email, and we will be happy to share our sequencing data.

Response 4: We have reviewed the article based on your suggestion, adding hyperlinks to the TAIR database and italicizing the gene names.

Response 5: We have revised the results and conclusion sections, and the changes can be seen in the revised manuscript.

Response 6: We have added statistical significance annotations below the figure legends.

Response 7: We have made revisions to the language and phrasing of the article.

---

## [Decision Letter · Decision Letter 1]

9 Mar 2025

BnLPAT2 Gene Regulates Oil Accumulation in Brassica napus

by Modulating Linoleic and Linolenic Acid Levels in Seeds

PONE-D-24-59454R1

Dear Dr. Xing,

We’re pleased to inform you that your manuscript has been judged scientifically suitable for publication and will be formally accepted for publication once it meets all outstanding technical requirements.

Kind regards,

Bahram Heidari

Academic Editor

PLOS ONE

Additional Editor Comments (optional):

Reviewers' comments:

Reviewer's Responses to Questions

**Comments to the Author**

1. If the authors have adequately addressed your comments raised in a previous round of review and you feel that this manuscript is now acceptable for publication, you may indicate that here to bypass the “Comments to the Author” section, enter your conflict of interest statement in the “Confidential to Editor” section, and submit your "Accept" recommendation.

Reviewer #1: All comments have been addressed

2. Is the manuscript technically sound, and do the data support the conclusions?

Reviewer #1: Yes

3. Has the statistical analysis been performed appropriately and rigorously? 

Reviewer #1: Yes

4. Have the authors made all data underlying the findings in their manuscript fully available?

Reviewer #1: Yes

5. Is the manuscript presented in an intelligible fashion and written in standard English?

Reviewer #1: Yes

6. Review Comments to the Author

Reviewer #1: (No Response)

7. PLOS authors have the option to publish the peer review history of their article (what does this mean? ). If published, this will include your full peer review and any attached files.

**Do you want your identity to be public for this peer review?** For information about this choice, including consent withdrawal, please see our Privacy Policy .

Reviewer #1: **Yes: ** Maryam Salami

---

## [Editor Report · Acceptance letter]

PONE-D-24-59454R1

PLOS ONE

Dear Dr. Xing,

I'm pleased to inform you that your manuscript has been deemed suitable for publication in PLOS ONE. Congratulations! Your manuscript is now being handed over to our production team.

Kind regards,

on behalf of

Dr. Bahram Heidari

Academic Editor

PLOS ONE